# Enhanced Desulfurization by Tannin Extract Absorption Assisted by Binuclear Sulfonated Phthalocyanine Cobalt Polymer: Performance and Mechanism

**DOI:** 10.3390/ma16062343

**Published:** 2023-03-15

**Authors:** Bing Wang, Huanyu Chen, Xingguang Hao, Kai Li, Xin Sun, Yuan Li, Ping Ning

**Affiliations:** 1Faculty of Environmental Science and Engineering, Kunming University of Science and Technology, Kunming 650500, China; 2Hunan Research Institute of Nonferrous Metals, Changsha 410100, China

**Keywords:** coke oven flue gas, flue gas desulfurization, tannin extract desulfurization, polymer

## Abstract

Removal of hydrogen sulfide (H_2_S) from coke oven gas has attracted increasing attention due to economic and environmental concerns. In this study, tannin extract (TE) absorption combined with binuclear sulfonated phthalocyanine cobalt organic polymer (OTS) and binuclear sulfonated phthalocyanine cobalt (PDS) with a fixed bed reactor is used for removal of H_2_S. The effect of gas flow rate, concentration of H_2_S, co-existence of organic sulfide compounds and O_2_ were investigated. Then, the effect of total alkalinity content of TE, NaVO_3_, OTS and PDS was studied in detail. The experimental results demonstrated that 100% H_2_S conversion could maintain for 13 h at a total alkalinity of 5.0 g/L, TE concentration of 4.0 g/L, NaVO_3_ concentration of 5 g/L, and OTS and PDS concentration of 0.2 g/L and 0.2 g/L, respectively. The OTS and PDS showed synergistic effect on boosting TE desulfurization efficiency. The results provide a new route for the investigation of liquid catalyzed oxidation desulfurization in an efficient and low-cost way.

## 1. Introduction

With the increase in the energy demand and rapid development of chemical industry and automobile consumption, the requirements for coke are constantly rising [1,2]. Coke is often produced by carbonization of coal in the steel industry. Coke oven gas (COG) is considered as a valuable by-product in the coking process, which contains ~55–60% H_2_, ~23–27% CH_4_, ~5–9% CO, ~1.9–4% CO_2_, ~3–5% N_2_, and ~0.4–0.8% O_2_ along with C_2_H_6_ and other hydrocarbons, H_2_S and NH_3_, in small quantities [3,4,5]. The COG can be utilized to produce high value-added chemical products such as hydrogen, methanol and synthetic natural gas. However, the existence of H_2_S may cause the poisoning of the catalysts used in the follow-up utilization of COG, such as the Ni-based catalysts. Additionally, H_2_S is also highly corrosive to steel equipment and poisonous to human beings. It is of great importance to remove H_2_S from COG for health and environmental concerns. The main technologies for H_2_S removal from COG in traditional coking industries include the Takahax method (TH) [6], Anthraquinone disulfonic acid method (ADA) [7,8], tannin extract absorption [9,10] and the PDS method [11]. Among them, tannin extract absorption and PDS method are promising methods due to their low cost and high efficiency.

Tannin extract is a water-soluble polyphenolic substance and a kind of polymer with molecular weights of 500–3000, of which the main component is tannins. Tannin is a mixed complex, containing numerous spilt hydroxy aromatic substances with a strong ability of oxygen adsorption. It also can act as a complexing agent to produce water-soluble vanadium complex with vanadium compounds. Tannins are extracted from the boiled stalks, leaves, and barks of plants [12,13]. In typical tannic extract technology (TE) for H_2_S removal, the H_2_S could be oxidated into elemental sulfur in the presence of an oxidant such as pentavalent vanadium. Gao et al. [14,15] have investigated the H_2_S absorption by pentavalent vanadium and tannic extract with oxidization state solution using cyclic voltammetry method. H_2_S could be absorbed to produce bisulfide ion HS^-^ and polysulfide ion S_x_^2−^. Then, these sulfur-containing substances were oxidized by pentavalent vanadium into sulfur, which could be easily separated from the liquid solution. However, the polysulfide ions S_x_^2−^ is also the main resource to generate byproducts such as thiocyanate, thiosulfate and sulfate due to its activity to react with oxygen [16,17]. These sulfur-containing compounds will cause the decline of the efficiency of desulfurization. In this regard, many efforts have been made to inhibit the production of the byproducts. Ji and co-workers [18] have conducted the research by using OTS as an additive in tannin extract absorption, which is binuclear sulfonated phthalocyanine cobalt organic polymer. The results showed that the addition of OTS could reduce the production of byproduct, where the OTS could promote the reaction of NaSCN to form NaHCO_3_. Nonetheless, the exhausted absorption solution was hardly to disposed. Equally importantly, the influence of the reaction condition on the desulfurization in tannin extract absorption is not clear, which would have great implications in an industrial application.

The PDS method is another promising technology for desulfurization which is widely used in Chinese industries, in which the binuclear sulfonated phthalocyanine cobalt is used as a catalyst to facilitate the oxidation of H_2_S [19]. The PDS method could prevent the reagent from HCN poisoning, and also possesses many advantages such as high sulfur capacity and low consumption of alkali and high oxidation speed because it can provide various double Co-N_4_ active sites to boost the reaction. However, the formation of S_x_^2−^ cannot be avoided, leading to the deep-oxidated products of sulfur containing salts. Indeed, it is urgent to find a new approach with high efficiency, low cost and less byproduct production.

Herein, we explore a novel method combining the improved tannic extract absorption with OTS and PDS methods for H_2_S removal. The PDS compound showed good capacity for O_2_ activation in the solution. The OTS polymer plays a vital role in the solubility of tannin extract. The NaVO_3_ is used first to replace V_2_O_5_ as an oxidant. Firstly, the reaction parameters including total alkalinity, content of tannin, Na_2_CO_3_, NaHCO_3_ and NaVO_3_ are thoroughly investigated to obtain an optimal reaction condition. Then, different additives including OTS, PDS, CuCl_2_, MnCl_2_, MgCl_2_, CaCl_2_, hydroquinone and picric acid are used for the comparative study. Multiple analytical technologies are employed to elucidate the reaction mechanism. Such acquired results could pave a new way for designing a flexible approach for H_2_S removal from industrial off-gas.

## 2. Experimental

### 2.1. Materials and Chemicals

Sodium carbonate (Na_2_CO_3_) and Sodium bicarbonate (NaHCO_3_) were purchased from Tianjin Chemical Reagent Co., Ltd., (Tianjin, China). Sodium metavanadate (NaVO_3_) was bought from Shanghai Macklin Biochemical Co., Ltd., (Shanghai, China). Tannin extract (TE), OTS and PDS were purchased from Guangxi Chemical Research Institute Co., Ltd., (Nanning, China). CuCl_2_, MgCl_2_, MnCl_2_, CaCl_2_, hydroquinone and picric acid were bought from Shanghai Aladdin Biochemical Technology Co., Ltd., (Shanghai, China). All reagents were used without further purification.

### 2.2. Batch Absorption Experiments

Absorption solution (200 mL) for H_2_S removal was prepared by taking different ratios of NaHCO_3_ and Na_2_CO_3_ to control the pH of the solution, and taking different masses of NaVO_3_ and TE in a 250 mL flask and placing them in a water bath with a magnetic stirrer at room temperature. The H_2_S from the gas cylinder was diluted with N_2_ (99.99%) to 300–1100 ppm, where the simulated gas stream total flow rate (Q) was fixed at 150–350 mL·min^−1^. The concentrations of H_2_S in the inlet and outlet gas from the reaction system were measured by a 9790 gas chromatograph (Jiangsu Fuli Analytical Instrument Co., Ltd., China) with a flame photometric detector (FTD) at 150 °C. In a typical batch experiment, 0.8 g Na_2_CO_3_, 0.8 g NaHCO_3_, 0.8 g TE and 0.6 g NaVO_3_ were added to 200 mL deionized water, and the Q was kept at 200 mL·min with 500 ppm H_2_S. The experimental equipment was shown in Appendix A.

### 2.3. Characterization Methods

The crystal phases were measured by the powder X-ray diffraction (XRD) patterns (Rigaku, Tokyo, Japan). The XRD systems were equipped with CuKα radiation (λ = 0.15406 nm) at a scanning rate of 1°/min in an angle of 2θ from 10° to 80°.

The valances of the surface element were recorded by X-ray photoelectron spectra (XPS) (ESCALAB 250Xi, Thermo Fisher Scientific) using Al Kα radiation. All the samples were calibrated using the C 1s peak of contaminant carbon of 284.8 eV as standard.

## 3. Results and Discussion

### 3.1. Optimization of Reaction Parameters

The TE technology is highly dependent on the operation condition, so we conducted investigation to optimize the reaction parameters. The gas flow rate, concentration of H_2_S, organic sulfur compound and content of oxygen was thoroughly investigated, as shown in Figure 1. From Figure 1b, the desulfurization rate could obtain 100% at the first 6 h for all gas flow rates. With the time prolonged, the desulfurization rate began to decrease first in a high flow rate. It may be because the higher flow rate makes the contact between H_2_S and NaVO_3_ quick. As a result, the time of consuming NaVO_3_ turns to be quick, followed by the deactivation of the solution. The effect of concentration of H_2_S has shown the same trend as the gas flow rate (Figure 1a). When H_2_S is 1100 ppm, the 100% removal rate could sustain for about 6 h. With the decreasing of concentration of H_2_S, the duration time of the 100% removal rate could sustain for up to 12 h at 300 ppm. In order to figure out the influence of organic sulfur compounds, we imported COS or CS_2_ separately with H_2_S (Figure 1c). It is worth noting that the COS or CS_2_ could be removed completely together with H_2_S at the first 8 h. However, after 8 h, the removal rate began to descend in the presence of COS or CS_2_, while the duration could sustain for 10 h without COS and CS_2_. It means that COS and CS_2_ could be transferred into H_2_S in the experimental condition, which will consume NaVO_3_, resulting in the reducing of the activity. The reaction was also investigated in the existence of 5% O_2_, as shown in Figure 1d. The reaction could be enhanced considerably in the presence of O_2_, and the duration of the 100% removal rate achieved 13 h, which is 3 h more than that of without O_2_. However, the byproducts will be produced afterwards. Additionally, the oxidation of multi-phenol structures into quinoid structures in TE could be completed by O_2_, where the quinoid structure plays a key role in the reaction from V^4+^ to V^5+^.

### 3.2. Effect of Total Alkalinity and Na_2_CO_3_ Content

The efficiency of TE technology was deeply influenced by the total alkalinity, which could be regulated by the dosage of NaHCO_3_. The impact of the amount of NaHCO_3_ was investigated, and the results were shown in Figure 2a. The obtained results showed that the duration increased with the increasing of the total alkalinity. When the total alkalinity was 0.2, the absorbent was exhausted quickly in 2 h. When the total alkalinity was 1.0, the 100% conversion could be kept for 7 h, which was much longer than that of the other total alkalinity. Then, the total alkalinity was kept at 1.0 and the amount of Na_2_CO_3_ was adjusted from 0.4 g to 1.2 g, where the amount of NaHCO_3_ was adjusted correspondingly. The efficiency of desulfurization was shown in Figure 2b. It can be seen that the performance in the H_2_S removal increased gradually with the increase in the amount of Na_2_CO_3_ when the total alkalinity is constant. The highest duration time of higher than 90% efficiency can be sustained for more than 8 h when the amount of Na_2_CO_3_ was 1.2 g. The reason could be attributed to the higher alkalinity of Na_2_CO_3_ than NaHCO_3_. The acidic gas H_2_S could react with the OH^−^ in the solution to form HS^−^ and S^2−^. In the same total alkalinity, more Na_2_CO_3_ could produce more OH^−^ ions, leading to higher H_2_S removal efficiency to form HS^−^ for further oxidation.

### 3.3. Effect of Content of TE

After optimizing the total alkalinity and content of Na_2_CO_3_, the effect of the content of TE was also investigated thoroughly, as shown in Figure 2c. The desulfurization duration was studied through varying the content of TE from 0.4 to 1.2 g. It could be observed that by varying the content of TE, the desulfurization duration could not be changed, of which the time was about 8 h. It can be concluded that the TE did not participate in the H_2_S oxidation and absorption directly. According to previous research [18], the product sodium divanadate of the reaction between NaVO_3_ and S^2−^ could be re-oxidated into NaVO_3_ in the presence of TE and O_2_. The reason was that quinoid structure in TE plays a key role in the oxidation process, while the multi-phenol structure has a negative effect on the oxidation which was the dominant species in TE. As a result, the content of TE in the anaerobic desulfurization exhibited a negligible effect on the H_2_S removal performance. It means that a small dosage of TE was enough for the highest activity. It was worth noting that the desulfurization duration could only sustain for 3.5 h without TE addition. The tannin extract plays a key role in complexing the vanadium compounds and the releasing of elemental sulfur. It means that the complexing effect with vanadium compounds to avoid the deposition of vanadium compounds in the water solution, which could extend the desulfurization activity.

### 3.4. Effect of Content of NaVO_3_

As NaVO_3_ is the main component in the TE technology, we further investigated the influence of the content of NaVO_3_ on the efficiency of H_2_S removal. The results were shown in Figure 2d. When the amount of NaVO_3_ was 0.2 g, the H_2_S could be completely removed by adsorbent in about 6 h. With the increasing of the amount of NaVO_3_, the duration of completed desulfurization extended gradually. When the amount of NaVO_3_ was 1.2 g, the duration had reached 10 h, which was about twice as high than that of 0.2 g. It was interesting to see that when the adsorbent began to lose activity, the speed of inactivation was fast, indicating that the activity of absorbent was positive correlated to the content of NaVO_3_. The high efficiency of high content of NaVO_3_ could be attributed to the existence of high valence state V^5+^, which has considerable oxidability. Moreover, the lattice oxygen in NaVO_3_ could directly take part in the oxidation of H_2_S into elemental sulfur, instead of the dissolved oxygen in the aqueous solution. In the reaction, V^5+^ had transferred into V^4+^ as well as the transformation from lattice oxygen to active oxygen species. In this regard, the reaction could be enhanced by increasing the content of NaVO_3_.

### 3.5. TE Technology Combined with OTS and PDS

In order to enhance the efficiency of TE method, we conducted the TE technology combined with OTS and PDS addition singly and simultaneously. Additionally, CuCl_2_, MnCl_2_, MgCl_2_, CaCl_2_, hydroquinone and picric acid were used as a single additive for comparison, as depicted in Figure 3a. It is clear to see that the inorganic metal salts and organic substances both demonstrate good catalytic performance in the TE process. They acted as the catalysts for the transformation from low valence V^4+^ to high valence V^5+^, resulting in the recovery of oxidants. The results showed that OTS has the best activity, while hydroquinone was the worst among those additives. The activity is in this order: OTS > PDS > MgCl_2_ = CaCl_2_ > picric acid > MnCl_2_ > CuCl_2_ > hydroquinone. The effect of OTS content was first investigated by adding different dosages of OTS into the TE system. The results were shown in Figure 3b. It is apparent that the addition of OTS could enhance the desulfurization efficiency to a large extent. Even a low amount of 0.05 g could prolong the duration from 8 h to 10 h. The longest duration of 100% efficiency was achieved by 0.1 g OTS, which was about 13.5 h. When a larger dosage of OTS than 0.1 g was added into the TE system, the activity began to descend, because the transition metal center could form a coordination compound with TE, which could boost the reaction synergistically, while excess additions will inhibit this effect. Meanwhile, the effect of PDS was also investigated separately, and the result was shown in Figure 3c. The addition of PDS demonstrated the same trend as OTS. It could achieve the highest activity at 0.1 g, then, the duration gradually descended with more OTS addition. Furthermore, the desulfurization efficiency was studied by adding both OTS and PDS. It can be seen from Figure 3d that the desulfurization efficiency could be enhanced by OTS or PDS addition only. When OTS and PDS were put into the system simultaneously, the efficiency was further boosted to about 13 h, and maintain about 100% conversion. It can be directly inferred that the co-addition of OTS and PDS has a synergistical positive effect on the desulfurization. The reason could be attributed to the different roles of OTS and PDS played in the tannin extract adsorption. When sulfonated salts such as OTS and PDS exist in the solution, the undissolved substances in the solution of tannin extract could be reduced. They could provide not only Co-N_4_ active sites, but also the C_4_ sites in the benzene ring which could coordinate with the ligands such as HS^−^ and S^2−^. When the OTS and PDS were added into the system simultaneously, a novel polymer was formed, leading to a higher conversation duration than adding singly. In this regard, OTS and PDS could supply considerable active sites for coordination with sulfur contained compounds. Importing the OTS and PDS into TE method, TE could be more dissolved and the catalytic activity could be enhanced due to the complexing effect of OTS polymer and PDS.

### 3.6. Insights into the Mechanism

From Figure 2, Na_2_CO_3_, NaHCO_3_, TE and NaVO_3_ play an important role in the removal of H_2_S when the tannin extract absorption method was used. In Figure 2d, it maintained for 2 h adsorbing H_2_S stably when NaVO_3_ was not added. It indicated that NaCO_3_ and NaHCO_3_ maintain the stability of the solution environment. From Figure 2c, the experimental results maintain for 8 h while increasing the mass of TE. It indicated that excessive TE was not necessary in the desulfurization process. To explore the H_2_S reaction mechanism, the characterization methods of XRD and XPS were used.

XPS analysis was performed on the samples to figure out the transfer and accumulation of surface sulfur element as time goes on.

Figure 4 presents the S 2p XPS spectra of the sample. Six different kinds of sulfur peaks in the spectrum were attributed to S 2p (Figure 4). The main surface species were $S_{P1/2}^{2-}$, $S_{P3/2}^{2-}$, V-S, element S (S^0^), S-O and S_n_ [20,21]. For $S_{P1/2}^{2-}$ and $S_{P3/2}^{2-}$, the peaks disappeared at 1 h, then appeared at 161.85 eV and 162.73 eV separately, followed by a shift to lower binding energy (BE) of 161.71 eV and a higher value of 162.93 eV. Lastly, it had the same tendency of lower BE of $S_{P1/2}^{2-}$ and higher $S_{P3/2}^{2-}$ [22,23,24]. It indicated that S^2−^ was transferred to other S species rapidly and the orbitals of S 2p were broken up (i.e., $S_{P3/2}^{2-}$ joined to chemical reaction while $S_{P1/2}^{2-}$ was steady and dispersed). As for the V-S, the peaks appeared at 163.64 eV and 163.44 eV at 1 h and 4 h separately [25,26]. Then, the peaks disappeared in the following experiments. It indicated that V-S formed stably due to the negligible low-shifting BE. The V transferred to the S to the active sites because of disappearing of V-S peaks. The S^0^ peaks always formed in the reaction process. It indicated that S^0^ was formed in the reaction. Over time, the BE displacement is 164.84 eV, 164.58 eV, 164.34 eV, 164.62 eV. It demonstrated that the products of S^0^ shift to the direction of low bounding energy with low stability in the reaction process. However, other products were formed to enhance the stability of S^0^ after 8 h. With regard to S-O, it appeared at BE of 168.26 eV, 168.24 eV, 167.94 eV and 167.59 eV the whole time. It indicated that S-O was unstable due to the low shift of BE in the process. It was clear to see that the peak area of S-O in the case of 8 h was much lower than at 7 h. It may be because the V transferred to the S to the active sites of O after V-S disappearing. As for the S_n_ peaks, it was formed after 7 h, and the BE were 167.94 eV and 168.74 eV. It indicated that the product of S_n_ was unsteady because of the reducing BE. Moreover, the formation of S_n_ was a benefit to the stabilization of S^0^.

Figure 5 presents the S 2p XPS spectra from the optimum amount of Na_2_CO_3_ at different times. It maintained the optimum total alkalinity. Compared with Figure 4a, the peaks of S^2−^ were appearing while V-S was disappearing in Figure 5a. It indicated that the reaction is dominated by the migration of sulfur ions with the increase in hydrolysis temperature in the solution. At 4 h (Figure 5b), the peak of S_n_ was formed while it was not in Figure 4b. It indicated that the reaction promotes the formation of product S with the increase in hydrolysis temperature in the solution. In Figure 5c, the peak areas of S_n_ and S^0^ were higher than that in Figure 4c. It indicated that the reaction promotes the formation of products of S and S_n_ with the increase in hydrolysis temperature in the solution.

Figure 6 presents the S 2p XPS spectra from the optimum amount of NaVO_3_ at different times. It maintained the optimum amount of Na_2_CO_3_. Compared with Figure 5a, the peaks of V-S were appearing while S^2−^ was disappearing in Figure 6a. It indicated that V-S was formed with the increasing mass of NaVO_3_. At 4 h (Figure 6b), the peak of S_n_ disappeared while it was formed in Figure 5b. It indicated that the reaction was not conducive to the formation of products of S_n_ in solution. In Figure 5d, the peak of V-S still exists. It indicated that V-S appeared in the whole reaction. It was hard to detect due to the low content of NaVO_3_.

The results of the fresh and exhausted residue after filtration of absorbent were studied to explain the details of the main reaction products and mechanisms. The XPS results indicated that H_2_S was oxide by NaVO_3_ to form the product of S_0_ and Sn. The XRD of NaVO_3_ was shown in Figure 7. The results show that the crystalline phases were NaVO_3_(PDF #75-0716). It was of an orthorhombic lattice type with a space group Pnma. As a result, the possible reaction mechanism is as followed:(1)$${\mathrm{Na}}_{2}{\mathrm{CO}}_{3}+H_{2}S\to \mathrm{NaHS}+{\mathrm{NaHCO}}_{3}$$
(2)$$2V^{5+}+{\mathrm{HS}}^{-}\to 2V^{4+}+H^{+}+S↓$$
(3)$$\mathrm{TQ}+V^{4+}+2H_{2}O\to \mathrm{YHQ}+V^{5+}+{\mathrm{OH}}^{-}\;$$
(4)$$\mathrm{TQ}+{\mathrm{HS}}^{-}\to \mathrm{YHQ}+S↓$$
where the quinoid structure of TE is denoted as TQ, and the multi-phenol structure of TE is denoted as YHQ. The addition of OTS or PDS will enhance the oxidation reaction (2), as they provided transition metal active centers for the oxidation of HS^−^ to elemental S. They take part in the redox cycle of V^5+^-V^4+^-V^5+^ as active catalysts. The PDS and OTS were binuclear sulfonated phthalocyanine cobalt compounds and its polymer, and they provided active Co-N_4_ sites to active O_2_. As a result, the V^4+^ could be easily oxidated back into the active phase of V^5+^.

In addition, the YHQ could be oxidated by O_2_ in the following regeneration process to return to TQ, forming H_2_O_2_ at the same time. The H_2_O_2_ could also oxidate the V^4+^ transferring to V^5+^, HS- into elemental sulfur. The reactions are as follows:(5)$$2\mathrm{YHQ}+O_{2}\to 2\mathrm{TQ}+H_{2}O_{2}$$
(6)$$H_{2}O_{2}+V^{4+}\to V^{5+}+2{\mathrm{OH}}^{-}$$
(7)$$H_{2}O_{2}+{\mathrm{HS}}^{-}\to H_{2}O+S↓+{\mathrm{OH}}^{-}$$

It is worth noting that further oxidated products such as S_2_O_3_^2−^ and S_2_O_4_^2−^ will be produced if the desulfurization solution contains NaHS in the presence of oxygen in the regeneration process. Therefore, the content of TE and NaVO_3_ are the key factors influencing the formation of byproducts, which should be controlled carefully.

## 4. Conclusions

In this study, the tannin extract technology combined with binuclear sulfonated phthalocyanine cobalt polymer (OTS) and compound (PDS) for H_2_S removal was thoroughly investigated. The reaction parameters including gas flow rate, concentration of H_2_S, co-existence of COS or CS_2_ and O_2_ were first optimized. High gas flow rate and concentration of H_2_S could lead to quick activity loss of desulfurization solution. The organic sulfur-containing compounds (COS and CS_2_) could be removed simultaneously. Then, the total alkalinity, the content of TE, NaVO_3_, OTS and PDS were studied in detail. The results demonstrate that the suitable conditions for the H_2_S removal are a total alkalinity of 5.0 g/L, TE concentration of 4.0 g/L, NaVO_3_ concentration of 5 g/L, and OTS and PDS concentrations of 0.2 g/L and 0.2 g/L, respectively. OTS could be beneficial for the solubility of TE. In addition, OTS and PDS have shown a catalytical effect on the redox cycles of V^5+^-V^4+^-V^5+^ and TQ-YHQ-TQ synergistically by supplying both Co-N_4_ and C_4_ in benzene ring active sites, which facilitates the H_2_S oxidation into elemental sulfur. This combined method provides new ideas for the development and research of desulfurization in industrial applications.

## Figures and Tables

**Figure 1 materials-16-02343-f001:**
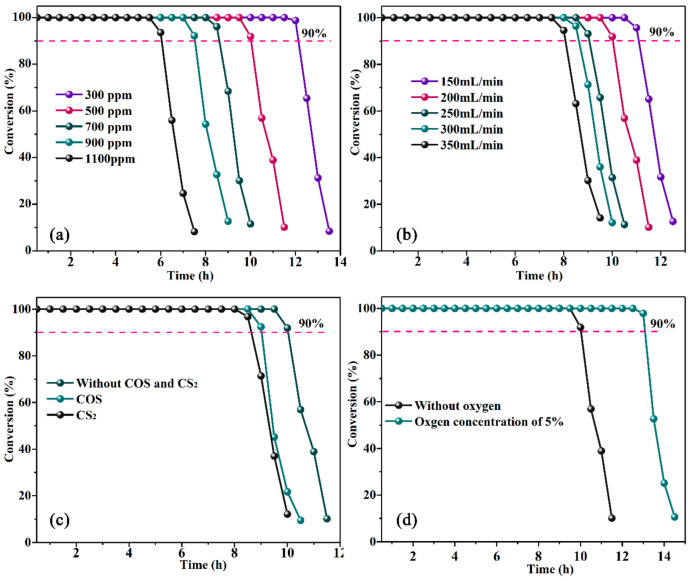
The H_2_S conversion in different reaction condition: (**a**) Gas flow rate = 150–350 mL/min. (**b**) H_2_S concentration = 300–1100 ppm. (**c**) with COS or CS_2_. (**d**) with or without 5% O_2_.

**Figure 2 materials-16-02343-f002:**
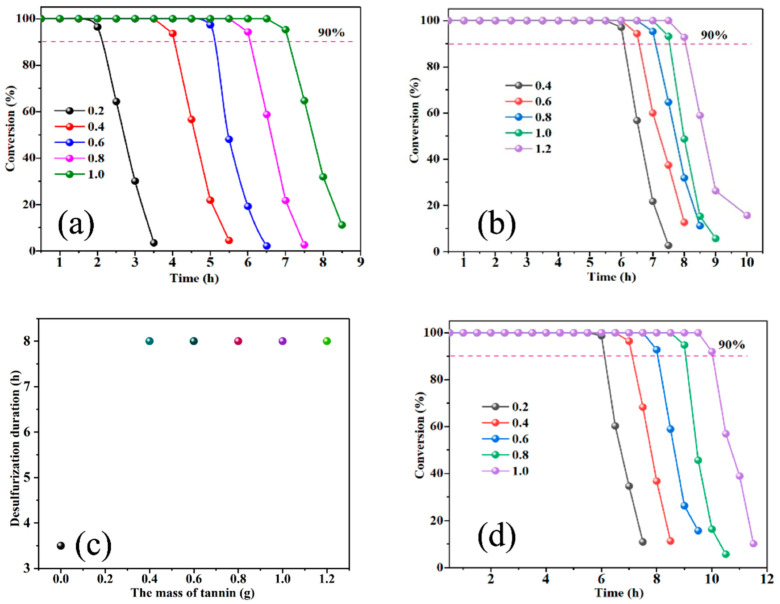
The conversion curve of H_2_S removal with various parameters: (**a**) the effect of total alkalinity from 0.2 to 1.0. (**b**) the effect of amount of Na_2_CO_3_: 0.4–1.2 g. (**c**) the effect of amount of TE: 0.4–1.2 g. (**d**) the effect of amount of NaVO_3_: 0.2–1.0 g.

**Figure 3 materials-16-02343-f003:**
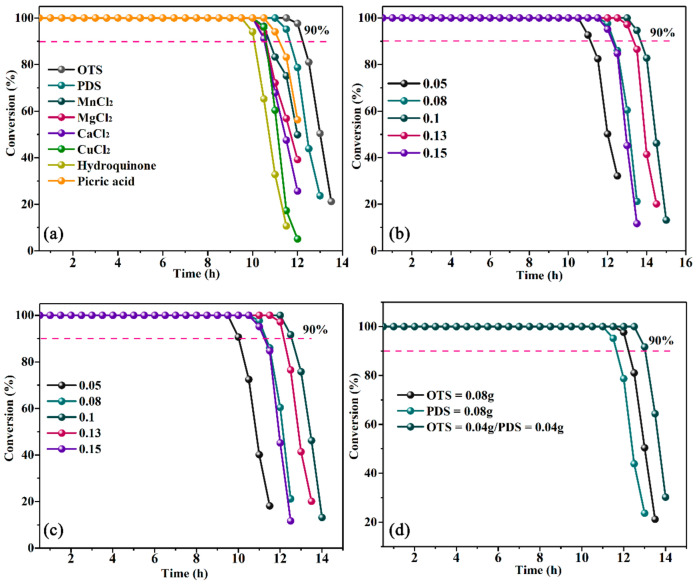
The effect of different additives in desulfurization efficiency: (**a**) various additive, (**b**) the effect of content of OTS: 0.05–0.15 g, (**c**) the effect of content of PDS: 0.05–0.15 g, (**d**) the effect of co-addition of OTS and PDS.

**Figure 4 materials-16-02343-f004:**
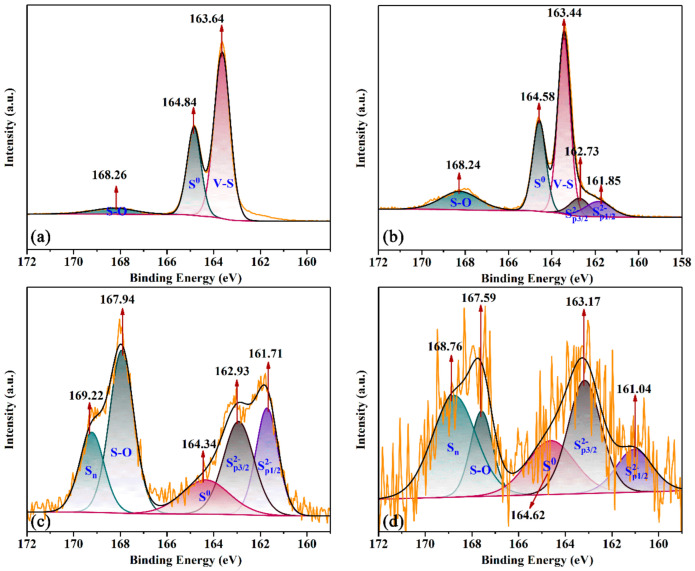
The XPS spectra from optimum total alkalinity at (**a**) 1 h; (**b**) 4 h; (**c**) 7 h; (**d**) 8 h.

**Figure 5 materials-16-02343-f005:**
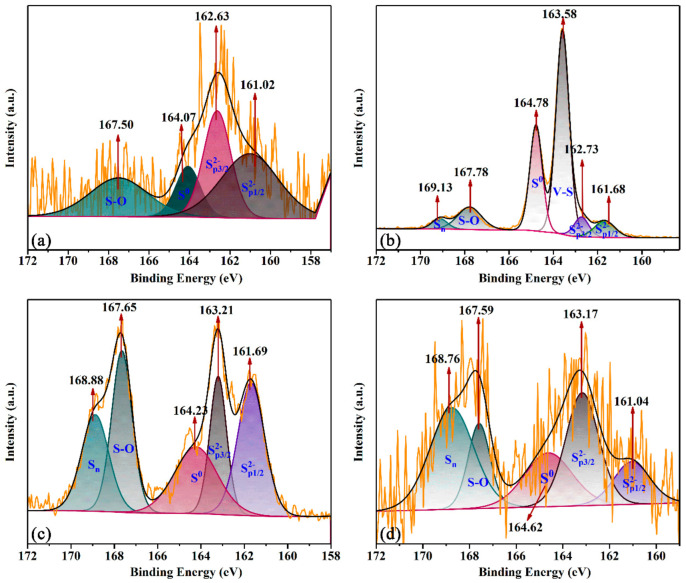
The XPS spectra from optimum amount of Na_2_CO_3_ at (**a**) 1 h; (**b**) 4 h; (**c**) 8 h; (**d**) 9 h.

**Figure 6 materials-16-02343-f006:**
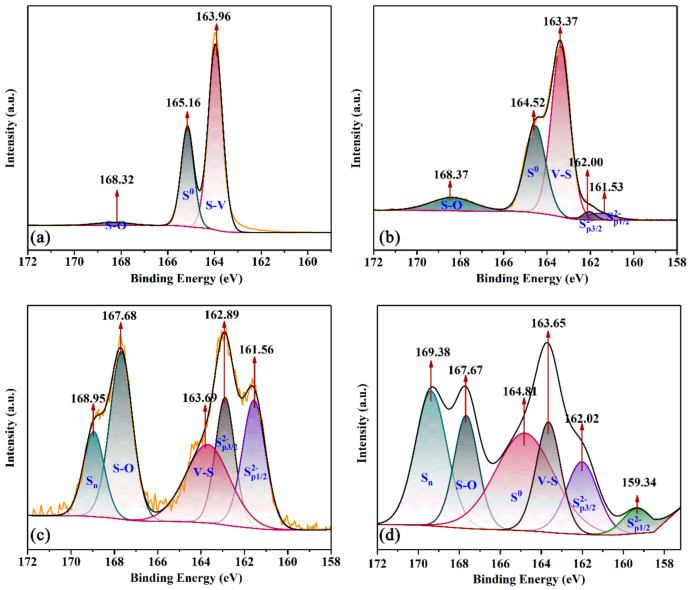
The XPS spectra from optimum amount of NaVO_3_ at (**a**) 1 h; (**b**) 4 h; (**c**) 10 h; (**d**) 11 h.

**Figure 7 materials-16-02343-f007:**
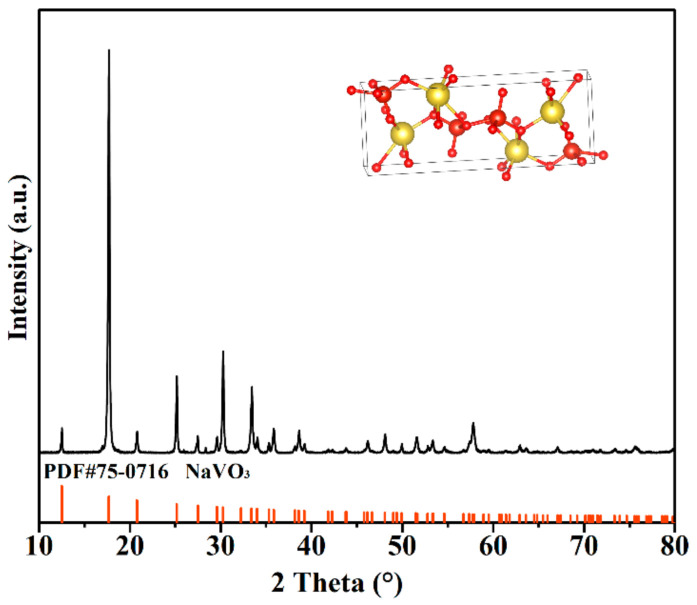
The XRD patterns of NaVO_3._

## Data Availability

The data presented in this study are available on request from the corresponding author. The data are not publicly available due to the data in this study involve enterprise production.

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
