# Peer review of "Enhanced Desulfurization by Tannin Extract Absorption Assisted by Binuclear Sulfonated Phthalocyanine Cobalt Polymer: Performance and Mechanism"

_materials, 2023, doi:10.3390/ma16062343_

Round 1

Reviewer 1 Report

In their paper, the authors describe a method for sulphur, specifically H2S, removal from flue gas. The method involves sodium carbonate/bicarbonate for pH control as well as sodium vanadate acting as oxidation catalyst in combination with oxygen. In addition tannin is added to the reaction mixture, but according to the experimental results this apparently has the role of a spectator in the reaction. The authors measure conversion/absorption rates of H2S using gas chromatography. The paper is based on a series of experiments with different experimental conditions (H2S concentration, gas flow rate, pH of the adsorbent solution, amounts of sodium vanadate and tannin were controlled parameters, duration until > 90% H2S breakthrough was the observable.

In my opinion this work is not suited to be published in "Polymers", as no polymer seems to be involved. A journal like "Processes" could be more suitable for this topic. In addition the paper can be improved by adding a control experiment without any tannin addition at all. The current results imply that tannin might be just a spectator, the additional experiment could give prove.
Further insight might also be gained by time resolved monitoring of the pH of the reaction solution. Maybe the eventual breakthrough of H2S is caused by a drop in pH. At least the correlation between breakthrough time and amount of base seems to indicate this could be a hidden variable in the system.
Would it also be possible to monitor formation of elementary sulfur in situ by e.g. using light-scattering methods?
The topic is definitely interesting and should be re-submitted to a different journal.

Author Response

Point 1: In their paper, the authors describe a method for sulphur, specifically H2S, removal from flue gas. The method involves sodium carbonate/bicarbonate for pH control as well as sodium vanadate acting as oxidation catalyst in combination with oxygen. In addition tannin is added to the reaction mixture, but according to the experimental results this apparently has the role of a spectator in the reaction. The authors measure conversion/absorption rates of H2S using gas chromatography. The paper is based on a series of experiments with different experimental conditions (H2S concentration, gas flow rate, pH of the adsorbent solution, amounts of sodium vanadate and tannin were controlled parameters, duration until > 90% H2S breakthrough was the observable.

In my opinion this work is not suited to be published in "Polymers", as no polymer seems to be involved. A journal like "Processes" could be more suitable for this topic. In addition the paper can be improved by adding a control experiment without any tannin addition at all. The current results imply that tannin might be just a spectator, the additional experiment could give prove.

Further insight might also be gained by time resolved monitoring of the pH of the reaction solution. Maybe the eventual breakthrough of H2S is caused by a drop in pH. At least the correlation between breakthrough time and amount of base seems to indicate this could be a hidden variable in the system.
Would it also be possible to monitor formation of elementary sulfur in situ by e.g. using light-scattering methods?

The topic is definitely interesting and should be re-submitted to a different journal.

Response 1:

Thanks for your suggestion.

Tannin extract is poly phenolic substances as a kind of high polymer, of which the main component is tannins. Tannin is a compound is a mixed complex, containing a lot spilt hydroxy aromatic substances with strong ability of adsorption oxygen. It also can act as a complexing agent to produce vanadium complex with vanadium compounds. The information was added into the text in line 39-42. Also, the OTS is an organic polymer compound. As a result, the enhanced effect is derived from the cooperation of TE, OTS and PDS, which is belong to the application of polymers. I think it is within the scope of the journal of “Ploymers”.

The pH plays a vital role in the desulfurization process because the first step of the reaction is H2S reacting with the HCO3- to form HS-. The decreasing of pH should lead to breakthrough of H2S eventually. We have conducted the experiments to investigate the effect of alkalinity of the solution.

It is hard to monitor formation of elementary sulfur in situ. We have analyzed the XPS curves of the exhausted catalysts, and we found that elemental sulfur and polysulfide were observed in them.

Reviewer 2 Report

This manuscript discusses the removal of H2S by combining tannin extract absorption with OTS and PDS. In this work, the reaction parameters are fully investigated, and the synergistic effect of OTS and PDS with tannin extract was also proposed. This work is valuable but needs to be properly improved before it is suitable for publication.

1.     The abbreviations should be explained in the first appearance, such as OTS, PDS, ADA and so on. Please check them in the whole text.

2.      Abstract: (1) the content of TE, NaVO3, OTS, PDS should be calculated in concentration instead of exact dosage. (2) The description of the synergistic effect of OTS and PDS is not clear, please add the related discussion.

3.     Fig. 2. (a), (b), (d) and Fig. 3. (b), (c) should improved the legends more clear where were only figures without descriptions. Furthermore, the colors of lines in fig. 3. (d) were similar, please make them more recognized.

4.     In section 3.6, “the addition of OTS or PDS will enhance the oxidation reaction (R2),”please explain it and add more description of its mechanisms.

5.     Line 286: “hydrolysis degree”, lind 286: “4h”,Line 289: “It indicated that the reaction promotes the formation of products of S and Sn with the increase of hydrolysis degree in solution.”, Line 309: “a orthorhombic”, the above words or usages may be wrong or hard to understand. The English of this paper should be greatly improved.

Author Response

This manuscript discusses the removal of H2S by combining tannin extract absorption with OTS and PDS. In this work, the reaction parameters are fully investigated, and the synergistic effect of OTS and PDS with tannin extract was also proposed. This work is valuable but needs to be properly improved before it is suitable for publication.

Point 1: The abbreviations should be explained in the first appearance, such as OTS, PDS, ADA and so on. Please check them in the whole text.

Response 1:

Thanks for your advice. The full names were added to the text in the first appearance.

OTS: binuclear phthalein cyanide sulfonate organic polymer compounds

PDS: binuclear sulfonated phthalocyanine cobalt

TH: Takahax

ADA: Anthraquinone disulfonic acid

Point 2: Abstract: (1) the content of TE, NaVO3, OTS, PDS should be calculated in concentration instead of exact dosage. (2) The description of the synergistic effect of OTS and PDS is not clear, please add the related discussion.

Response 2: Thanks for your advice, (1) we have modified the content into concentration in line 17-18 : otal alkalinity of 5.0 g/L, TE concentration of 4.0 g/L, NaVO3 concentration of 5 g/L, OTS and PDS concentration of 0.2 g/L and 0.2 g/L.

(2) we have added “The OTS and PDS were the binuclear sulfonated phthalocyanine cobalt compounds and its polymer, and they provide active Co-N4 sites to active O2. As a result, the V4+ could be easily oxidated back into the active phase of V5+.” Into line 280.

Point 3: Fig. 2. (a), (b), (d) and Fig. 3. (b), (c) should improved the legends more clear where were only figures without descriptions. Furthermore, the colors of lines in fig. 3. (d) were similar, please make them more recognized.

Response 3: The figures were modified.

Point 4:  In section 3.6, “the addition of OTS or PDS will enhance the oxidation reaction (R2),”please explain it and add more description of its mechanisms.

Response 4: we have added “The OTS and PDS were the binuclear sulfonated phthalocyanine cobalt compounds and its polymer, and they provide active Co-N4 sites to active O2. As a result, the V4+ could be easily oxidated back into the active phase of V5+.” Into line 280.

Point 5: Line 286: “hydrolysis degree”, lind 286: “4h”,Line 289: “It indicated that the reaction promotes the formation of products of S and Sn with the increase of hydrolysis degree in solution.”, Line 309: “a orthorhombic”, the above words or usages may be wrong or hard to understand. The English of this paper should be greatly improved.

Response 5: The sentences related to “hydrolysis degree” were deleted.

Reviewer 3 Report

The manuscript is well prepared and the results are well discussed. Please consider the following minor comments:

Line 34: please state the acronyms' full terms when they firstly appear (eg: TH, ADA, PDS). Please check other acronyms as well (OTS, NaVO3, etc).

Line 150 - 156: "According to previous research...... negligible effect on the H2S removal performance". Please cite the references involved.

Figure 2(c), the symbol+line plot is suggested to replaced with symbol only.

Author Response

The manuscript is well prepared and the results are well discussed. Please consider the following minor comments:

Point 1:Line 34: please state the acronyms' full terms when they firstly appear (eg: TH, ADA, PDS). Please check other acronyms as well (OTS, NaVO3, etc).

Response 1:

Thanks for your advice. The full names were added to the text in the first appearance.

OTS: binuclear phthalein cyanide sulfonate organic polymer compounds

PDS: binuclear sulfonated phthalocyanine cobalt

TH: Takahax

ADA: Anthraquinone disulfonic acid

Point 2: Line 150 - 156: "According to previous research...... negligible effect on the H2S removal performance". Please cite the references involved.

Response 2: 

The reference is reference [19], which is added.

Figure 2(c), the symbol+line plot is suggested to replaced with symbol only.

The figure is modified.

Round 2

Reviewer 1 Report

In the updated version of the manuscript a week link to the polymeric nature of tannin was added. However the manuscript is still lacking evidence that tannin plays any role in the target reaction studied. Figure 2c seems to prove that tannin has no effect at all and a control experiment without any tannin in the reaction mixture (TE = 0) to prove the TE-effect is still missing. I still have the opinion this paper is not about a polymer reactant and should not be published in this journal but leave this decision to the editor. 

Author Response

Point 1: In the updated version of the manuscript a week link to the polymeric nature of tannin was added. However the manuscript is still lacking evidence that tannin plays any role in the target reaction studied. Figure 2c seems to prove that tannin has no effect at all and a control experiment without any tannin in the reaction mixture (TE = 0) to prove the TE-effect is still missing. I still have the opinion this paper is not about a polymer reactant and should not be published in this journal but leave this decision to the editor.

Response 1:

Thanks for your advice. We have conducted the experiment without TE addition. The results showed that the desulfurization activity could sustain above 90% for only 3.5 hours without tannin extract addition, as shown in Fig. 2 (c), which is shown at below.

Tannin extract is water-soluble polyphenolic substance as a kind of polymer with molecular weights of 500-3000, of which the main component is tannins. The tannin extract plays a key role on the complexing the vanadium compounds and the releasing of elemental sulfur. It means that the complexing effect with vanadium compounds to avoid the deposition of vanadium compounds in the water solution which could extend the desulfurization activity. In addition, binuclear sulfonated phthalocyanine cobalt organic polymer (OTS) is a kind of organo-metallic cobalt coordination polymer with complex structure, providing various Co-N4 active sites for the H2S oxidation. The transition metal Co centers could coordinate with fifth ligand and sixth ligand such as SH- and S2- to boost the electron transfer. As well, the OTS could form novel polymer with PDS to enhance the processing, because the activity of adding both OTS and PDS are higher than adding OTS and PDS. Although the OTS and PDS had good affinity towards SH- and S2-, the affinity was too strong to release the product elemental sulfur. In this regard, the tannin extract could be helpful to resolve the problem. We also conducted the experiments that using OTS and PDS as the additive singly and simultaneously without tannin extract. The results showed that the conversion above 90% could sustain about 5.8 h when the dosage of OTS and PDS were 0.04 g and 0.04 g, respectively, shown in the figure below. The results are much lower than that with TE addition. It could reveal the vital effect of TE in the system. Furthermore, the price of tannin extract is much lower than OTS, which has financial benefits in the industrial application.

Reviewer 2 Report

I acknowledge that the paper was improved with appropriate changes, and therefore I recommend its publication at this stage.

Reviewer 3 Report

The manuscript is recommended for acceptance.